# Lifestyles and their relative contribution to biological aging across multiple-organ systems: Change analysis from the China Multi-Ethnic Cohort study

Yuan Zhang[1†], Dan Tang[1,2†], Ning Zhang[1], Yi Xiang[1], Yifan Hu[1], Wen Qian[3], Yangji Baima[4], Xianbin Ding[5], Ziyun Wang[6], Jianzhong Yin[7]*, Xiong Xiao[1]*, Xing Zhao[1]

[1]West China School of Public Health and West China Fourth Hospital, Sichuan University, Chengdu, China; [2]Xiamen Center for Disease Control and Prevention, Xiamen, China; [3]Chengdu Center for Disease Control and Prevention, Chengdu, China; [4]School of Medicine, Tibet University, Lhasa, China; [5]Chongqing Municipal Centre for Disease Control and Prevention, Chongqing, China; [6]School of Public Health, the Key Laboratory of Environmental Pollution Monitoring and Disease Control, Ministry of Education, Guizhou Medical University, Guiyang, China; [7]School of Public Health, Kunming Medical University, Kunming, China

*For correspondence:
yinjianzhong2005@sina.com (JY);
xiaoxiong.scu@scu.edu.cn (XX)

†These authors contributed equally to this work

## eLife Assessment

This **useful** study examined the associations of a healthy lifestyle with comprehensive and organ-specific biological ages defined using common blood biomarkers and body measures. Its large sample size, longitudinal design, and robust statistical analysis provide **solid** support for the findings, which will be of interest to epidemiologists and clinicians.

## Abstract

**Background:** Biological aging exhibits heterogeneity across multi-organ systems. However, it remains unclear how is lifestyle associated with overall and organ-specific aging and which factors contribute most in Southwest China.

**Methods:** This study involved 8396 participants who completed two surveys from the China Multi-Ethnic Cohort (CMEC) study. The healthy lifestyle index (HLI) was developed using five lifestyle factors: smoking, alcohol, diet, exercise, and sleep. The comprehensive and organ-specific biological ages (BAs) were calculated using the Klemera–Doubal method based on longitudinal clinical laboratory measurements, and validation were conducted to select BA reflecting related diseases. Fixed effects model was used to examine the associations between HLI or its components and the acceleration of validated BAs. We further evaluated the relative contribution of lifestyle components to comprehension and organ systems BAs using quantile G-computation.

**Results:** About two-thirds of participants changed HLI scores between surveys. After validation, three organ-specific BAs (the cardiopulmonary, metabolic, and liver BAs) were identified as reflective of specific diseases and included in further analyses with the comprehensive BA. The health alterations in HLI showed a protective association with the acceleration of all BAs, with a mean shift of –0.19 (95% CI –0.34, –0.03) in the comprehensive BA acceleration. Diet and smoking were the major

contributors to overall negative associations of five lifestyle factors, with the comprehensive BA and metabolic BA accounting for 24% and 55% respectively.

**Conclusions:** Healthy lifestyle changes were inversely related to comprehensive and organ-specific biological aging in Southwest China, with diet and smoking contributing most to comprehensive and metabolic BA separately. Our findings highlight the potential of lifestyle interventions to decelerate aging and identify intervention targets to limit organ-specific aging in less-developed regions.

**Funding:** This work was primarily supported by the National Natural Science Foundation of China (Grant No. 82273740) and Sichuan Science and Technology Program (Natural Science Foundation of Sichuan Province, Grant No. 2024NSFSC0552). The CMEC study was funded by the National Key Research and Development Program of China (Grant No. 2017YFC0907305, 2017YFC0907300). The sponsors had no role in the design, analysis, interpretation, or writing of this article.

## Introduction

Aging is a global issue, with about 10% of the world population aged 65 years and over in 2022, and the proportion expected to be near 16% by 2050 (*Department of Economic and Social Affairs and ECONOMIC UNDF, 2023*). In China, the proportion has reached 14.2% at the end of 2021 (*Pot et al., 2022*). Aging leads to reduced life expectancy and chronic diseases through dynamic and heterogeneous changes in biological systems (*Kennedy et al., 2014*). While chronological age (CA) is widely used as a marker of aging, biological age (BA) has been constructed as a more accurate indicator for biological aging (*Hamczyk et al., 2020*). Aging exhibits variations across and within individuals; people with the same CA can age at different rates, and distinct organ systems within an individual can also age differently (*Schaum et al., 2020*). Comprehensive aging indicators like epigenetic clocks and PhenoAge, capturing different aging aspects, have been developed (*Jylhävä et al., 2017*; *Li et al., 2020*). Recent studies have focused on organ-specific aging markers to understand their aging rate variations (*Nie et al., 2022*; *Tian et al., 2023*). The process of biological aging, influenced by genetics, environment, and health behaviors, is modifiable, yet research has predominantly concentrated on comprehensive BA (*Bauer and Newman, 2022*; *Belsky et al., 2015*; *Elliott et al., 2021*; *Fiorito et al., 2021*). Nevertheless, current research on determinants of multi-organ biological aging remains limited. Further investigation is crucial to deepen understanding of comprehensive and organ-specific biological aging and to develop targeted interventions for age-related diseases of each organ system.

Lifestyle factors such as smoking, drinking, diet, physical activity, and sleep are recognized as contributors to many age-related diseases (*Han et al., 2021*; *Zhang et al., 2021*; *Henry et al., 2019*), and researchers have begun to explore how lifestyle factors influence the process of biological aging. Studies on lifestyle and BA can be summarized into two categories, one focusing on the association of a single lifestyle factor with biological aging (*Klopack et al., 2022*; *Kresovich et al., 2021*; *Kresovich et al., 2022*; *Gao et al., 2022*) because lifestyle factors may be interrelated and have cumulative effects, the other further focusing on the impact of a combined lifestyle on biological aging. Research focusing on healthy lifestyles has been conducted in Chinese (*Wang et al., 2022*; *Liang et al., 2023*), American (*Thomas et al., 2023*), and British populations (*Yang et al., 2022*) and has found that adherence to a healthy lifestyle is consistently associated with slower biological aging. However, most studies used one-time measurements of lifestyle factors as exposure and could not examine the impact of lifestyle changes on BA. In addition, research has often focused on epigenetic age and PhenoAge. The Klemera–Doubal method of biological age (KDM-BA) has been shown to perform well in predicting frailty and mortality and is used to identify the BAs of organ systems (*Nie et al., 2022*; *Zhong et al., 2020*). Therefore, KDM-BA, based on clinical lab data sets, offers accessible and cost-effective means for aging identification and intervention. Besides, no study has been conducted to analyze further the relative contributions of multiple lifestyle changes on various organ systems BA, which could identify important influences on the comprehensive and organ-specific biological aging to prioritize interventions, especially in developing regions with limited resources. Southwest China is characterized by multi-ethnicity, unbalanced internal development, and rapid aging, making identifying priority intervenable aging factors important.

Therefore, based on the China Multi-Ethnic Cohort (CMEC) study, this study aimed to develop potential BAs for organ systems using the KDM algorithm, explore how lifestyle changes correlate

**eLife digest** Everyone ages, but some people show signs of aging faster than others. There are also important differences between chronological aging, which measures how many years a person has lived, and biological aging, which involves the biological processes that cause age-related changes in the body. Some people may have slower biological aging than others despite being the same chronological age. Genetics, the environment, and lifestyle differences likely explain why some people have slower biological aging than others.

Large studies assessing both lifestyle changes and biological aging may provide new insights into aging. One such study is the China Multi-Ethnic Cohort Study, which followed about 100,000 people in Southwest China between 2018 and 2021. The study included a diverse sample of people from different ethnic backgrounds and collected lifestyle information and data on various organ systems.

Zhang, Tang et al. show that diet improvements and quitting smoking can slow biological and organ-specific aging. The analysis looked at data from about 8,400 participants in the China Multi-Ethnic Cohort Study who completed two surveys about their lifestyles over the course of the study. Laboratory measurements from the participants assessed biological aging in various organ systems over the same period. Adopting a healthy diet like the Mediterranean diet had the greatest effect on overall biological aging, contributing 24% to the overall protective effect of healthier lifestyle modifications. Quitting smoking had the greatest impact on metabolic health, responsible for 55% of the slowdown in metabolic aging seen with healthier lifestyle changes. The changes were particularly pronounced in more disadvantaged groups who may disproportionately face environmental stress or have more limited access to resources.

The analysis shows that healthy lifestyle changes can slow biological aging, particularly in regions with fewer health resources. The results suggest that targeted lifestyle interventions may help delay age-related conditions like heart or liver disease in such areas. It also suggests that promoting healthy diets and quitting smoking are the most impactful lifestyle changes. More studies are needed to confirm the results and test interventions that overcome barriers to healthy lifestyles in under-resourced areas.

with both comprehensive and organ-specific BAs, and further investigate the relative contributions of individual lifestyle components in southwest China.

## Methods
### Study population

This study used data from the CMEC baseline and repeated surveys for analysis. The CMEC adopted a multi-stage stratified cluster sampling, with participants covering five provinces and seven ethnic groups (Tibetan, Yi, Miao, Bai, Bouyei, Dong, and Han) in Southwest China, representing the local regional and ethnic characteristics. The baseline survey of the CMEC was conducted between May 2018 and September 2019, and the repeated survey was conducted between August 2020 and June 2021. The CMEC baseline survey included 99,556 individuals, with 10% of the sample population selected for the repeated survey. Details of the CMEC procedures and methodology have been described in our previous study (*Zhao et al., 2021*). Participants' personal interviews, physical examinations, and clinical laboratory measurements were completed using a uniform standardized operating procedure in both surveys. The study protocol was approved by the Sichuan University Medical Ethical Review Board (ID: K2016038, K2020022). All participants in the study signed an informed consent form before the investigation.

In this study, we included participants with available information on lifestyle, measurements for calculating BA required, and covariates. We excluded individuals with missing data to calculate BA and individuals with incomplete data on lifestyle and covariates for participants at the baseline and repeated survey. Finally, 8396 individuals aged 30–79 were included as the full set in the longitudinal analysis after matching (*Figure 1*).

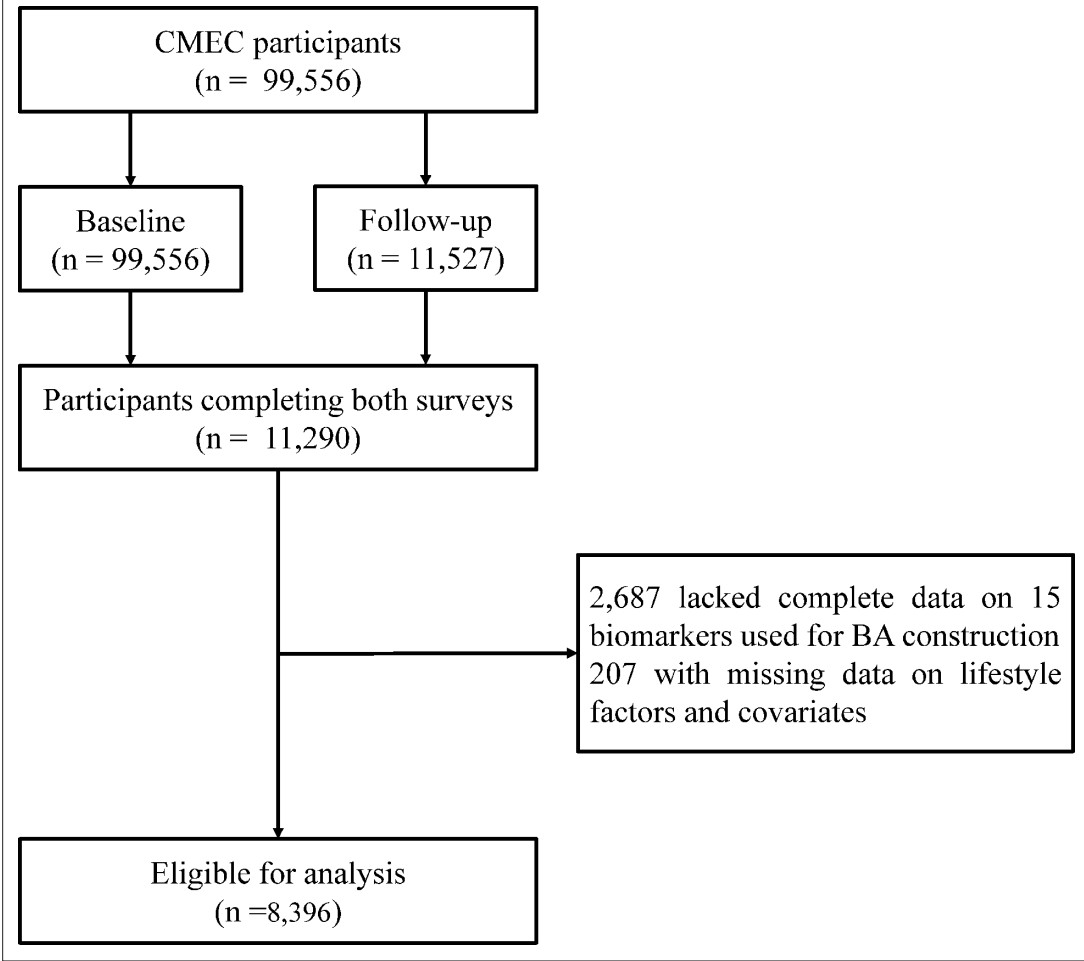

**Figure 1.** Flowchart of the study.

## Definition of lifestyle factors

Considering previous studies (*Zhang et al., 2021*; *Guasch-Ferré et al., 2022*), the healthy lifestyle index (HLI) was established through five lifestyle factors: smoking, alcohol consumption, diet, exercise, and sleep. Data for the lifestyle factors were obtained from the questionnaire, and the simplified semi-quantitative food frequency questionnaire (FFQ) was assessed by trained staff. The FFQ's reproducibility and validity were evaluated by conducting repeated FFQs and 24-hour dietary recalls (*Xiao et al., 2021*).

The definitions of a healthy lifestyle are as follows. Smoking was categorized according to smoking status, and never smoking was defined as healthy. For alcohol consumption, according to the findings of recent research on alcohol consumption in Chinese populations, being healthy was defined as being a non-regular drinker (drinking frequency less than once a week) (*Millwood et al., 2023*). For diet, since the Mediterranean diet (MED) is currently the most internationally recognized healthy dietary pattern and has widely and robustly beneficial effects (*Mazza et al., 2021*), we used the alternative Mediterranean diet (aMED) to evaluate dietary quality, with details shown in the supplemental material. Since the HLI already contained a drinking component, we removed the drinking item in the aMED, which had a score range of 7–35. We defined individuals with aMED scores ≥population median as healthy diets. More details on the calculation of aMED can be found in previous studies (*Xiao et al., 2021*). For exercise, based on how often participants participated in physical activity in their leisure time during the past year, regular exercise ('1–2 times/week', '3–5 times/week', or 'daily or almost every day') was categorized as healthy (*Luo et al., 2022*). For sleep, a sleep duration of 7–8 hours was defined as healthy sleep based on previous studies (*Chaput et al., 2020*).

In addition, to comprehensively evaluate the impact of overall healthy lifestyle level on outcomes, we further constructed the HLI in this study. For each lifestyle, we assign 1 point to health and 0 points to the opposite. The HLI was calculated by directly adding up the five lifestyle scores, ranging from 0 to 5, with a higher score representing an overall healthier lifestyle, denoted as HLI (range) in the following text. We then transformed HLI into a dichotomous variable in this study, denoted as HLI (category), where a score of 4–5 for HLI was considered a healthy lifestyle, and a score of 0–3 was considered an unfavorable lifestyle that could be improved.

## Covariate assessment

The selection of covariates is based on prior literature review and previously constructed directed acyclic graphs. Covariates included age, sex, ethnicity (majority, minority), urbanicity (rural, urban), education (no schooling, primary school, middle/high school, college/university), occupation (primary industry practitioner, secondary industry practitioner, tertiary industry practitioner, unemployed), marital status (married/cohabiting, not married/cohabiting), total energy intake (kcal/day), depression symptoms, anxiety symptoms, menopausal status in women (premenopausal, perimenopausal, postmenopausal), beverage intake (never, former consumer, currently consuming), dietary supplement intake, self-reported doctor-diagnosed diseases of diabetes, cardiovascular disease (CVD), and cancer. Among these, sex, ethnicity, urbanicity, and education were considered time-invariant variables, while other variables were considered time-varying.

## BA construction

The comprehensive BA and BAs across multiple-organ systems were estimated using the Klemera–Doubal method (*Klemera and Doubal, 2006*), which is well-validated in the Chinese population (*Chen et al., 2023*). We selected indicators for constructing BA from clinical lab data sets measured in the baseline and repeated surveys of the CMEC, filtering based on a missing rate of less than 30%. Based on previous studies, these indicators were categorized into five systems based on the organ/system function they represent: cardiopulmonary, metabolic, liver, renal, and immune systems, as detailed in *Supplementary file 1b* (*Nie et al., 2022*, *Tian et al., 2023*). Thus, we developed a comprehensive BA alongside five organ system-specific BAs: cardiopulmonary BA, metabolic BA, liver BA, renal BA, and immune BA. The selection process was then completed following the methodology used in our previous studies, which are detailed in the supplementary materials.

After finishing the screening process, 15 measures were used in constructing the comprehensive BA, which were systolic blood pressure (SBP), waist-to-hip ratio (WHR), peak expiratory flow (PEF), γ-glutamyl transpeptidase (GGT), albumin (ALB), low-density lipoprotein cholesterol (LDL-CH), high-density lipoprotein cholesterol (HDL-CH), triglyceride (TG), aspartate aminotransferase (AST), creatinine (Cr), alkaline phosphatase (ALP), urea, mean corpuscular volume (MCV), glycosylated hemoglobin (HBA1C), and platelet count (PLT). The cardiopulmonary BA was assessed using SBP and PEF; the metabolic BA was assessed using LDL-CH, HDL-CH, HBA1C, TG, and WHR; the liver BA was assessed using AST, GGT, ALP, and ALB; the renal BA was assessed using Cr and urea; and the immune BA was assessed using PLT and MCV. Using the above measures, we calculated each BA in the male and female populations separately using the KDM algorithm. To quantify variation in biological aging between participants, we calculated BA acceleration, which is the difference between each BA and CA simultaneously.

## BA validation

To assess the performance of the newly developed BAs of multiple-organ systems, we performed a validation analysis to evaluate the relationships between them and organ-specific diseases. Given the short follow-up period in the CMEC study, we conducted a cross-sectional analysis using baseline data. We separately examined the associations between the comprehension BA and CVD, diabetes, cancer, cardiopulmonary BA and CVD and chronic bronchitis, the metabolic BA and CVD and diabetes, the liver BA and chronic hepatitis or cirrhosis, the immune BA and rheumatoid arthritis. For interpretability, we simultaneously standardized and classified each BA acceleration into categories (|BA acceleration|≤1, BA acceleration < –1, and BA acceleration >1). We used logistic regression to assess the associations between each continuous and categorized BA (|BA acceleration|≤1 serving as the reference group) and the diseases. Since the values of BA acceleration of the cardiopulmonary BA

were small, it was not processed as categorical variables. The model was additionally adjusted for five healthy lifestyle factors based on the covariates. The comprehensive BA constructed in the CMEC study has been validated to better reflect age-related disease and frailty (*Xiang et al., 2024*). After completing the validation analysis, we selected BAs reflecting aging and related diseases for further analysis.

## Statistical analysis

We described changes in lifestyle between the two waves of surveys, where changes were categorized as becoming healthier, remaining unchanged, and becoming unhealthier. For single lifestyles, remaining unchanged indicates that the factor has remained healthy or unhealthy at two surveys; becoming healthier suggests that it was unhealthy at baseline and became healthy on the repeated survey, and vice versa. For the HLI scores, no change indicates the same score in both surveys, a change to healthier implies a repeated survey score higher than the baseline score, and a shift in less healthy indicates a repeated survey score lower than the baseline score. The dichotomous HLI's change categorization was similar to the single lifestyle. The baseline characteristics of participants were described across categories in the dichotomized HLI, with continuous variables represented by median (interquartile range) and categorical variables by proportions (*Table 1*).

We used the fixed effects model (FEM) to examine the associations between each lifestyle factor and the acceleration of validated BAs for people who participated in both surveys. FEM is extensively applied in the analysis of panel data in the fields of sociology, economics, and public health research, where it serves to control for unobserved individual heterogeneity. The detailed methodology of the FEM is outlined in the supplementary materials. Because of the approximately 2-year interval between the two CMEC waves, we assumed that the short-term effect of lifestyle changes on biological aging was linear during this period. Multivariable-adjusted models were constructed to account for potential confounding, including time-varying and time-invariant variables and baseline CA. We incorporated five lifestyle factors as exposures simultaneously into the model, while the continuous and binary HLI were entered into the model as exposures separately. Since the BAs we analyzed may potentially measure overlapping aspects of human aging, we did not correct for multiple comparisons.

To further obtain the relative contribution of each lifestyle factor on validated comprehension and organ systems BA, the present study used data from two waves for quantile G-computation (QGC) analysis. QGC is a statistical technique for evaluating the effects of mixture exposures, capable of discerning positive and negative influences, and is widely used in environmental epidemiology research (*Keil et al., 2020*; *Zhang et al., 2024*). Additionally, subgroup analyses for continuous and categorized HLI and each validated BA were conducted across sex (male vs. female), baseline age (<60 vs. ≥60), ethnicity (majority vs. minority), urbanicity (rural vs. urban), baseline BA acceleration (<0 vs. ≥0), and baseline disease status (free of diabetes, CVD, and cancer vs. either one). The heterogeneity between strata was assessed using the Q test ($\alpha = 0.1$ was considered significant heterogeneity).

We performed sensitivity analyses concerning exposure definitions, confounders, and our analysis method. First, we repeated the analysis of the association between lifestyles and BAs using a standard FEM, with adjustments made only for time-varying variables. Second, we used alternative common health criteria for each lifestyle factor separately, and we repeated the FEM analyses and QGC analyses by replacing the health definition of one lifestyle factor at a time. Exposure definitions for the main analyses and sensitivity analyses are shown in *Supplementary file 1a*. Third, we did not adjust for body mass index (BMI) in models because some studies have suggested that BMI may be an intermediate factor between lifestyles and health outcomes. However, in the sensitivity analysis, we additionally adjusted for BMI in the FEM and QGC analyses models. Statistical analyses were performed using R Project for Statistical Computing version 4.1.1 (Vienna, Austria).

## Results

### Lifestyle changes between two waves in CMEC

*Figure 2* shows the changes in lifestyle factors in the study population from baseline to repeated surveys. For the HLI (range), more than 60% of participants experienced a lifestyle change. For the HLI (category), about 30% of participants changed lifestyle categories, and the percentages were close for both types. For each lifestyle factor, the proportion of individuals who remained unchanged during

**Table 1.** Characteristics of participants in baseline and repeated surveys, based on the categorized HLI*.

| Characteristic | Baseline HLI (category) | | Follow up HLI (category) | |
|---|---|---|---|---|
| | Unfavorable N=4935 | Healthy N=3461 | Unfavorable N=5121 | Healthy N=3275 |
| Age (years) | 51.83 [44.97, 60.44] | 49.19 [42.51, 56.66] | 53.75 [47.01, 62.24] | 50.86 [44.21, 59.14] |
| Female (%) | 2511 (50.9) | 2666 (77.0) | 2685 (52.4) | 2492 (76.1) |
| Urbanicity (%) | 1472 (29.8) | 1525 (44.1) | 1527 (29.8) | 1470 (44.9) |
| Majority (%) | 2665 (54.0) | 2427 (70.1) | 2763 (54.0) | 2329 (71.1) |
| Education (%) | | | | |
| No schooling | 1506 (30.5) | 653 (18.9) | 1554 (30.3) | 605 (18.5) |
| Primary school | 1305 (26.4) | 721 (20.8) | 1377 (26.9) | 649 (19.8) |
| Middle/high school | 1764 (35.7) | 1525 (44.1) | 1780 (34.8) | 1509 (46.1) |
| College/university | 360 (7.3) | 562 (16.2) | 410 (8.0) | 512 (15.6) |
| Occupation (%) | | | | |
| Primary industry practitioner | 1993 (40.4) | 820 (23.7) | 2093 (40.9) | 826 (25.2) |
| Secondary industry practitioner | 335 (6.8) | 197 (5.7) | 396 (7.7) | 174 (5.3) |
| Tertiary industry practitioner | 1699 (34.4) | 1671 (48.3) | 1736 (33.9) | 1462 (44.6) |
| Unemployed | 908 (18.4) | 773 (22.3) | 896 (17.5) | 813 (24.8) |
| Married (%) | 4396 (89.1) | 3111 (89.9) | 4595 (89.7) | 2966 (90.6) |
| Total energy intake (kcal/day) | 1796.35 [1394.49, 2293.37] | 1759.71 [1404.92, 2156.58] | 1510.72 [1129.17, 1994.64] | 1482.29 [1183.94, 1879.82] |
| BMI (kg/m$^2$) | 24.29 [21.94, 26.79] | 24.12 [22.04, 26.47] | 24.61 [22.23, 27.05] | 24.34 [22.31, 26.73] |
| Depression symptom (%) | 289 (5.9) | 117 (3.4) | 203 (4.0) | 86 (2.6) |
| Anxiety symptom (%) | 335 (6.8) | 146 (4.2) | 212 (4.1) | 71 (2.2) |
| Menopausal status in women (%) | | | | |
| Premenopausal | 1042 (41.5) | 1336 (50.1) | 914 (34.0) | 1099 (44.1) |
| Perimenopausal | 170 (6.8) | 189 (7.1) | 133 (5.0) | 109 (4.4) |
| Postmenopausal | 1299 (51.7) | 1141 (42.8) | 1638 (61.0) | 1284 (51.5) |
| Beverage (%) | | | | |
| Never | 4564 (92.5) | 3292 (95.1) | 4746 (92.7) | 3079 (94.0) |
| Former consumer | 23 (0.5) | 16 (0.5) | 27 (0.5) | 14 (0.4) |
| Currently consuming | 348 (7.1) | 153 (4.4) | 348 (6.8) | 182 (5.6) |
| Dietary supplement intake (%) | 711 (14.4) | 722 (20.9) | 724 (14.1) | 712 (21.7) |
| Major diseases | | | | |
| Diabetes (%) | 245 (5.0) | 161 (4.7) | 275 (5.4) | 159 (4.9) |
| CVD (%) | 974 (19.7) | 628 (18.1) | 1054 (20.6) | 602 (18.4) |
| Cancer (%) | 36 (0.7) | 31 (0.9) | 39 (0.8) | 32 (1.0) |
| Biological ages[†] | | | | |
| Comprehensive BA | 52.35 [43.90, 61.07] | 49.15 [40.19, 58.61] | 54.78 [46.48, 63.45] | 51.67 [42.74, 61.31] |
| Comprehensive BA acceleration | 0.00 [−2.89, 3.00] | −0.70 [−3.79, 2.73] | 0.40 [−2.66, 3.48] | −0.13 [−3.21, 3.25] |
| Cardiopulmonary BA | 51.86 [45.05, 60.40] | 49.10 [42.48, 56.89] | 53.79 [47.07, 62.14] | 50.82 [43.97, 59.18] |
| Cardiopulmonary BA acceleration | 0.08 [−0.47, 0.59] | −0.03 [−0.65, 0.55] | 0.04 [−0.56, 0.62] | −0.08 [−0.71, 0.56] |

*Table 1 continued on next page*

*Table 1 continued*

| Characteristic | Baseline HLI (category) | | Follow up HLI (category) | |
|---|---|---|---|---|
| | Unfavorable N=4935 | Healthy N=3461 | Unfavorable N=5121 | Healthy N=3275 |
| Metabolic BA | 52.87 [43.86, 61.78] | 49.42 [40.53, 59.01] | 55.14 [46.10, 64.29] | 52.29 [42.89, 61.77] |
| Metabolic BA acceleration | 0.55 [−3.86, 4.85] | −0.37 [−4.41, 3.86] | 0.46 [−3.87, 4.94] | 0.32 [−3.93, 4.46] |
| Liver BA | 52.42 [41.08, 63.96] | 49.40 [37.83, 60.43] | 57.61 [46.15, 68.52] | 54.13 [42.31, 64.32] |
| Liver BA acceleration | 0.15 [−8.16, 8.46] | −0.88 [−8.50, 6.73] | 2.88 [−5.31, 11.58] | 1.11 [−6.02, 9.08] |

BA, biological age; BMI, body mass index; CVD, cardiovascular disease; HLI, healthy lifestyle indicator.

*Data are presented as median (25th, 75th percentile) for continuous variables and count (percentage) for categorical variables. For HLI (category), 'healthy' corresponds to a score of 4–5, while 'unfavorable' corresponds to a score of 0–3.

†Information on each validated BA has been reported. BA acceleration is the difference between each BA and CA in the same survey.

the two surveys was higher than that of individuals who changed, with smoking and drinking having the highest proportion of unchanged individuals at more than 90%, indicating that these two lifestyle factors are less likely to change. Sleep and diet behaviors had a relatively high percentage of change, accounting for over one-third, and exercise had a slightly lower rate of change, nearly 30%.

## Construction and validation of organ systems BAs

Details of the comprehensive and organ systems BAs are presented in *Supplementary file 1c*. The results of the cross-sectional analysis between the comprehensive and organ systems BAs and diseases are presented in *Supplementary file 1d*. Accelerated comprehensive BA was positively correlated with an increased risk of CVD and diabetes. The cardiopulmonary BA, the liver BA, and the metabolic BA were each linked to an increased risk of their respective diseases. An increase in cardiopulmonary

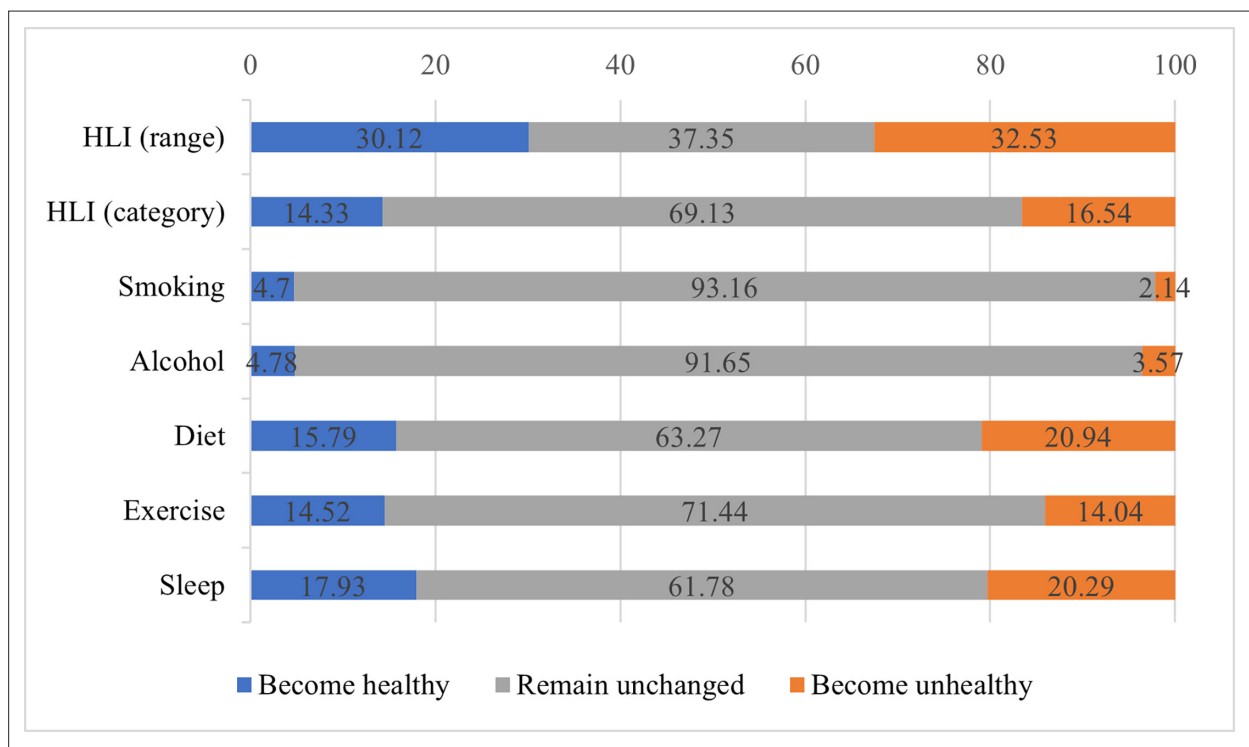

**Figure 2.** Changes in lifestyle factors from baseline to repeated survey. The blue bars represent the percentage of individuals who have transitioned to healthier behaviors or improved their healthy lifestyle indicator (HLI) score; the gray bars represent the percentage of individuals whose behaviors or HLI score have not changed; the orange bars represent the percentage of individuals who have transitioned to unhealthier behaviors or whose HLI score has decreased.

**Table 2.** Associations of healthy lifestyle factors and HLI with the BA acceleration of validated BAs.

| Variables | Comprehensive BA acceleration β (95% CI) | Cardiopulmonary BA acceleration β (95% CI) | Metabolic BA acceleration β (95% CI) | Liver BA acceleration β (95% CI) |
|---|---|---|---|---|
| HLI (range) | | | | |
| Per 1-point increase | –0.11 (–0.19, –0.03) | –0.02 (–0.04, –0.01) | –0.12 (–0.22, –0.02) | –0.23 (–0.47, 0.00) |
| HLI (category) | | | | |
| To have a healthy lifestyle | –0.19 (–0.34, –0.03) | –0.03 (–0.07, 0.00) | –0.16 (–0.36, 0.03) | –0.19 (–0.66, 0.27) |
| Healthy lifestyle factors | | | | |
| ΔSmoking | –0.13 (–0.47, 0.21) | –0.03 (–0.10, 0.04) | –0.54 (–0.97, –0.11) | –0.57 (–1.58, 0.44) |
| ΔAlcohol | –0.17 (-0.47, 0.13) | –0.02 (–0.09, 0.04) | –0.18 (–0.56, 0.21) | –0.59 (–1.50, 0.31) |
| ΔDiet | –0.15 (–0.29, –0.00) | –0.01 (–0.04, 0.02) | –0.18 (–0.36, 0.00) | –0.10 (–0.53, 0.33) |
| ΔExercise | –0.16 (–0.32, 0.00) | –0.03 (–0.07, 0.00) | –0.09 (–0.30, 0.11) | –0.05 (–0.53, 0.44) |
| ΔSleep | –0.02 (–0.16, 0.12) | –0.02 (–0.05, 0.01) | 0.01 (–0.17, 0.19) | –0.36 (–0.78, 0.06) |

Estimates were obtained using FEMs treating the BA accelerations as the dependent variables and HLI (as either continuous or as categorized) or five individual lifestyle factors as the independent variables. Models were adjusted for age, occupation, marital status, total energy intake, depression symptoms, anxiety symptoms, menopausal status in women, beverage intake, dietary supplement intake, diabetes, cardiovascular disease, cancer, sex, ethnicity, urbanicity, education, and the participants' age at baseline.

BA, biological age; HLI, healthy lifestyle indicator. ΔSmoking, change in smoking status between the baseline and repeated survey; ΔAlcohol, change in alcohol consumption between the baseline and repeated survey; ΔDiet, change in dietary quality between the baseline and repeated survey; ΔExercise, change in exercise between the baseline and repeated survey; ΔSleep, change in sleep between the baseline and repeated survey.

BA acceleration was associated with a higher risk of CVDs but not with chronic obstructive pulmonary disease. Per 1 SD increase of the liver BA was associated with a higher risk of chronic hepatitis or cirrhosis (OR 1.28, 95% CI 1.23, 1.33), and per 1-SD increase in metabolic BA corresponded to a higher risk of CVD (OR 1.21, 95% CI 1.19, 1.23) and diabetes (OR 3.23, 95% CI 3.12, 3.34). However, no expected associations with related diseases were observed for immune BA. Finally, we concluded that the cardiopulmonary BA, the metabolic BA, and the liver BA could reflect organ-specific disease and were included in subsequent analyses along with comprehensive BA.

## Characteristics of participants by HLI categories

*Table 1* reports the characteristics of populations categorized by dichotomous HLI at baseline and the repeated survey. In this study, 3461 individuals had a healthy lifestyle, and individuals with an unfavorable lifestyle accounted for a higher proportion (n [%] = 4935 [58.8%]) at baseline. There was a slight decrease in the proportion of people with a healthy lifestyle at the time of the repeated survey (n [%] = 3275 [39.0%]), while 5049 (61.0%) had an unfavorable lifestyle.

Since the lifestyle categories of participants may change, the distribution of population characteristics between baseline and repeated surveys changes slightly but is generally similar. Participants with healthy lifestyles were more likely to be female, in urban areas, Han Chinese, with higher levels of education, less likely to be employed in primary and secondary industries, more likely to be premenopausal and with dietary supplement intake, less likely to have anxiety, depression, and younger in BA, as shown by the lower median of BA and BA acceleration of the comprehensive, cardiopulmonary, metabolic, and liver BA. For major diseases, the proportion of people with a healthier lifestyle was closer to that of people with an unfavorable lifestyle in both surveys.

## Associations of lifestyle factors and HLI changes with BA acceleration and relative contributions

*Table 2* shows the associations of individual lifestyle factors and continuous and dichotomous HLI with all validated BAs acceleration. For comprehensive BA, a significant negative association was found between HLI and accelerated biological aging, with a mean change of –0.19 (95% CI –0.34,–0.03) in the BA acceleration for a healthier lifestyle change. Results showed that a change of –0.15 (95% CI

−0.29, −0.00) in the comprehensive BA acceleration was associated with a shift in diet. For the cardio-pulmonary BA, metabolic BA, and liver BA, the HLI was associated with a reduction in each BA acceleration, although some of the results were not statistically significant. Shift in smoking was related to a change in metabolic BA acceleration of –0.54 (95% CI –0.97, –0.11), and the results of individual lifestyle factors and the cardiopulmonary BA as well as the liver BA did not reach statistical significance.

The relative contributions of each lifestyle factor on the comprehensive and the cardiopulmonary BA, the metabolic BA, and the liver BA acceleration based on the QGC method are shown in *Figure 3*. For comprehensive BA, the relative contribution of diet was 0.24. Although the relative contributions of drinking and exercise were slightly higher than that of diet, they were not statistically significant. For metabolic BA, smoking was the major contributor, weighting 0.55. For cardiopulmonary BA and liver BA, exercise and alcohol consumption were the most contributing components in all lifestyle factors separately, although the results were statistically significant.

## Results of subgroup analysis and sensitivity analysis

*Figure 4* shows the results of the subgroup analyses of the HLI range and category with the comprehensive BA acceleration. For the HLI range, the direction of the association was consistent with the whole population, with statistical significance only in females, the younger, minorities, rural residents, accelerated BA group, and no baseline disease populations. The negative association was stronger in minorities (hterogeneity test p=0.028). For categorized HLI and AA, the direction of the association was broadly consistent with the whole population. The continued HLI results for cardiopulmonary BA showed a stronger negative association in females (heterogeneity test p=0.045, *Figure 4—figure supplement 1*). For metabolic BA and liver BA, no differences were observed between different subgroups (*Figure 4—figure supplements 2 and 3*).

We conducted a series of sensitivity analyses to test the robustness of the results. Firstly, the results of the standard FEM analysis were broadly consistent with the primary analysis (*Supplementary file 1e* and *Figure 3—figure supplement 1*). For the comprehensive BA, exercise, along with diet, was negatively correlated with BA acceleration, and in the main analysis, exercise was borderline statistically significant. While the results of some lifestyles for liver function BA changed, none were statistically significant. Secondly, changing the definition of health for any lifestyle factor had negative association estimates with all BAs acceleration for the HLI. Although the effect size of some lifestyle factors changed, most of the results remained stable. Full results are reported in *Supplementary file 1f* and *Figure 3—figure supplements 2–5*. Thirdly, with additional adjustment for BMI, the direction and magnitude of the associations were generally consistent with the main analysis results (*Supplementary file 1g* and *Figure 3—figure supplement 6*). In summary, the sensitivity analysis results aligned with the main results and were relatively robust.

## Discussion

This study examined the association of lifestyle changes on biological aging in the Southwest Cohort based on two surveys. About two-thirds of the participants were observed to change their HLI scores between the two surveys. The health alterations in HLI showed a protective association between accelerated aging of comprehension and each organ system. The relative contribution of diet was the largest among all lifestyle components for the comprehensive BA, and smoking contributed the most to the metabolic BA, respectively, suggesting that these factors are particularly crucial in specifically decelerating the pace of biological aging.

In our study, the prevalence of healthy lifestyles was higher than the prevalence reported by research based on the China Kadoorie Biobank study and the China Nutrition and Health Surveillance (*Sun et al., 2022*), with those with 4–5 lifestyle factors accounting for 41.2 and 39.0% at baseline and repeated surveys, respectively. The relatively high prevalence of healthy lifestyles in this study may be explained by the recent establishment of the CMEC and an overall trend toward healthier lifestyles, alongside different lifestyle standards. Furthermore, we noted lifestyle changes in our research and subsequently employed the FEM for analysis based on this observation. Lifestyle may be influenced by environmental factors such as urbanization and individual characteristics such as health awareness, and thus, lifestyle may change over time, yet fewer studies have focused on this (*Lao et al., 2014*). The high proportion of changes occurring in sleep, exercise, and diet here suggests that they may be

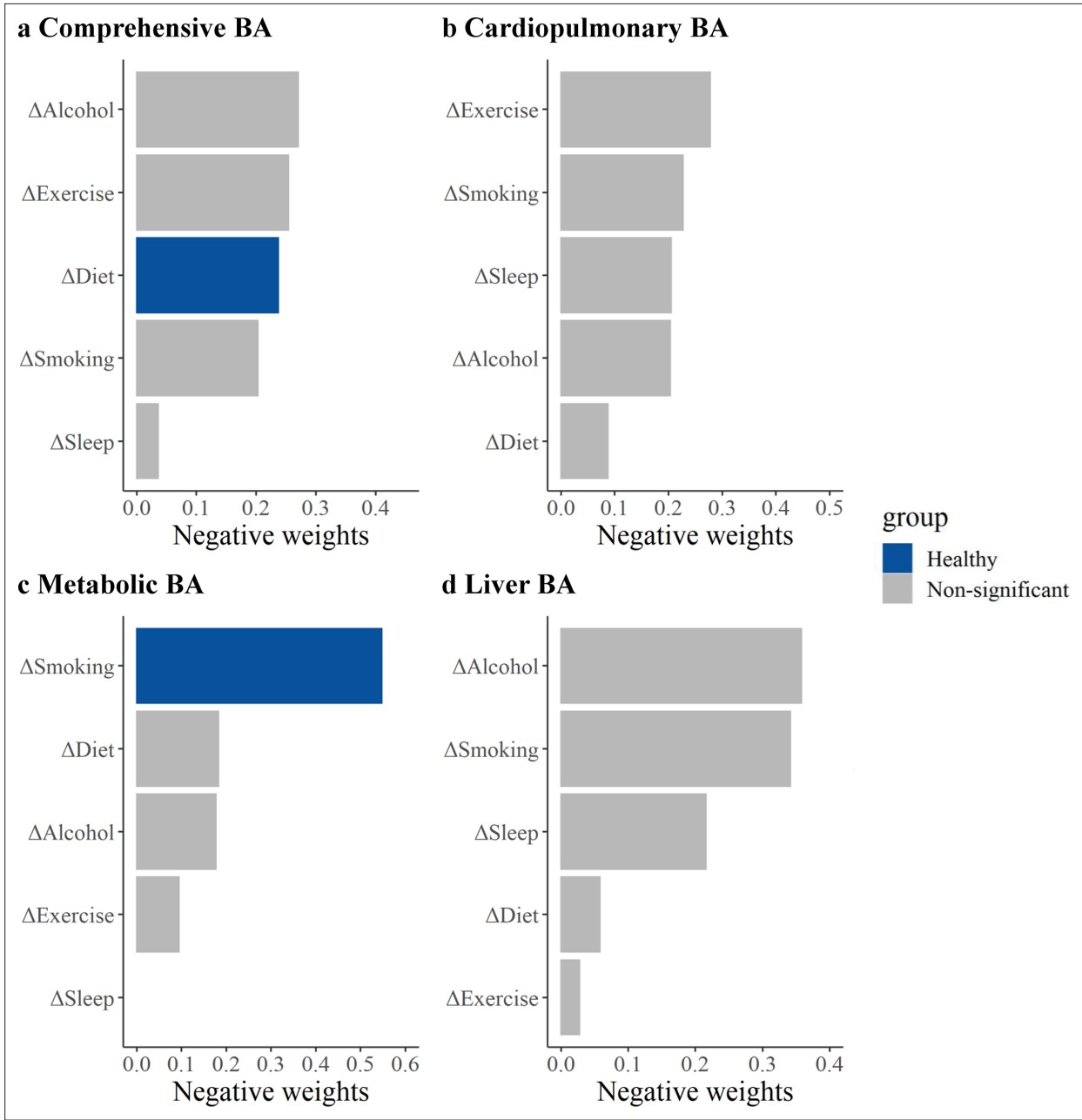

**Figure 3.** Relative contributions of five healthy lifestyle components to biological age (BA) acceleration. Panels: results of the comprehensive BA acceleration, the cardiopulmonary BA acceleration, the metabolic BA acceleration, and the liver BA acceleration (**a–d**). ΔSmoking, change in smoking status between the baseline and repeated survey; ΔAlcohol, change in alcohol consumption between the baseline and repeated survey; ΔDiet, change in dietary quality between the baseline and repeated survey; ΔExercise, change in exercise between the baseline and repeated survey; ΔSleep, change in sleep between the baseline and repeated survey. Estimates were obtained using quantile G-computation (QGC), which treated the BA accelerations as the dependent variables and five individual lifestyle factors as the independent variables. Models were adjusted for age, occupation, marital status, total energy intake, depression symptoms, anxiety symptoms, menopausal status in women, beverage intake, dietary supplement intake, diabetes, cardiovascular disease, cancer, sex, ethnicity, urbanicity, education, and the participants' age at baseline. The blue bars represent results that are statistically significant in the fixed effects model (FEM) analysis, while the gray bars represent results in the FEM analysis that were not found to be statistically significant and positive weights were not shown.

The online version of this article includes the following figure supplement(s) for figure 3:

**Figure supplement 1.** Relative contributions of five healthy lifestyle components to biological age (BA) acceleration with adjustment for time-varying covariate.

*Figure 3 continued on next page*

*Figure 3 continued*

**Figure supplement 2.** The relative contributions of five healthy lifestyle components to the comprehensive biological age (BA) acceleration with altered healthy lifestyle criteria.

**Figure supplement 3.** The relative contributions of five healthy lifestyle components to the cardiopulmonary biological age (BA) acceleration with altered healthy lifestyle criteria.

**Figure supplement 4.** The relative contributions of five healthy lifestyle components to the metabolic biological age (BA) acceleration with altered healthy lifestyle criteria.

**Figure supplement 5.** The relative contributions of five healthy lifestyle components to the liver biological age (BA) acceleration with altered healthy lifestyle criteria.

**Figure supplement 6.** Relative contributions of five healthy lifestyle components to the comprehensive and the metabolic biological age (BA) acceleration with additionally adjusted for body mass index (BMI).

more variable and relatively easy to intervene. For overall health styles, there was a higher proportion of changes in lifestyle scores but fewer shifts from unhealthy to healthy or healthy to unhealthy. That is, while lifestyle fluctuates over time, it is less likely to change substantially, consistent with the findings of another study (*Han et al., 2021*; *Zhou et al., 2021*).

Our study reveals that healthy lifestyle shifts can slow the pace of comprehensive BA, with diet being a crucial component. Both the HLI range and category showed robust protective associations. Although there are differences in lifestyle definitions and aging indicators, available studies consistently find that healthier lifestyles are associated with lower aging accelerationv (*Wang et al., 2022*; *Thomas et al., 2023*; *Kim et al., 2022*; *Simons et al., 2022*). Despite limited research on how lifestyle components contribute to aging, our study provides additional evidence. We used the widely recommended aMED as a healthy dietary criterion, which includes rich plant foods, olive oil, moderate fish, and red wine, some of which may slow aging and promote health (*Wang et al., 2023*). In our study, the association between exercise and comprehensive BA was nearly statistically significant. Higher levels of exercise have been shown to be associated with a lower incidence of chronic disease and a longer lifespan (*Watts et al., 2022*; *Kankaanpää et al., 2021*). Diet and exercise may slow aging through beneficial DNA methylation changes (*Galkin et al., 2023*), modification of gut microbiota (*Zhang et al., 2022*), and reducing oxidative stress and inflammation (*Aleksandrova et al., 2021*; *El Assar et al., 2022*), collectively enhancing fitness and slowing the aging pace. Combining the previous randomized controlled trial with our findings, the evidence suggested that diet and exercise are easily modifiable anti-aging factors in the real world (*Fiorito et al., 2021*).

Another contribution of our study is presenting evidence of different lifestyle factors influencing multi-organ BA, which was constructed and validated for Southwest China. The HLI still played a protective role against aging across various organ systems, yet the components contributing most varied specifically. Biological aging varies across organ systems, and the associations with lifestyle may differ. Change in smoking played a significant role in metabolic aging. The current study of the British population has also identified smoking as a factor influencing aging across multiple-organ systems (*Tian et al., 2023*). The Coronary Artery Risk Development in Young Adults Study revealed that smoking contributed to epigenetic aging acceleration by 83.5% (*Kim et al., 2022*). The variation in results could be attributed to the fact that current studies offer evidence of cumulative exposure to smoking (*Klopack et al., 2022*; *Wang et al., 2022*), while our research focuses on short-term changes. Short-term smoking behavior changes may be sufficient to impact metabolism aging but do not significantly affect the comprehensive BA. Existing research has shown that the metabolic and immune systems can impact various systems through inter-organ aging pathways, making it essential to focus on modifiable factors of metabolic system aging. Hence, we discovered that smoking could serve as an intervention target for organ system aging, holding the potential to delay the onset of diseases in specific systems, thereby extending a healthy lifespan.

In subgroup analysis, we discovered that negative associations of HLI and the comprehensive BA acceleration varied among ethnic groups. No observed differences in HLI category change between ethnic groups, potentially due to insufficient statistical power. Compared to the majority group, the minority may have lower socioeconomic status and less health awareness, leading to less management of lifestyle factors, and, consequently, stronger associations when changes in these factors occur. Given that biological aging is a cause of various long-term outcomes, these results highlight the importance of conducting interventions among minority groups in Southwest China.

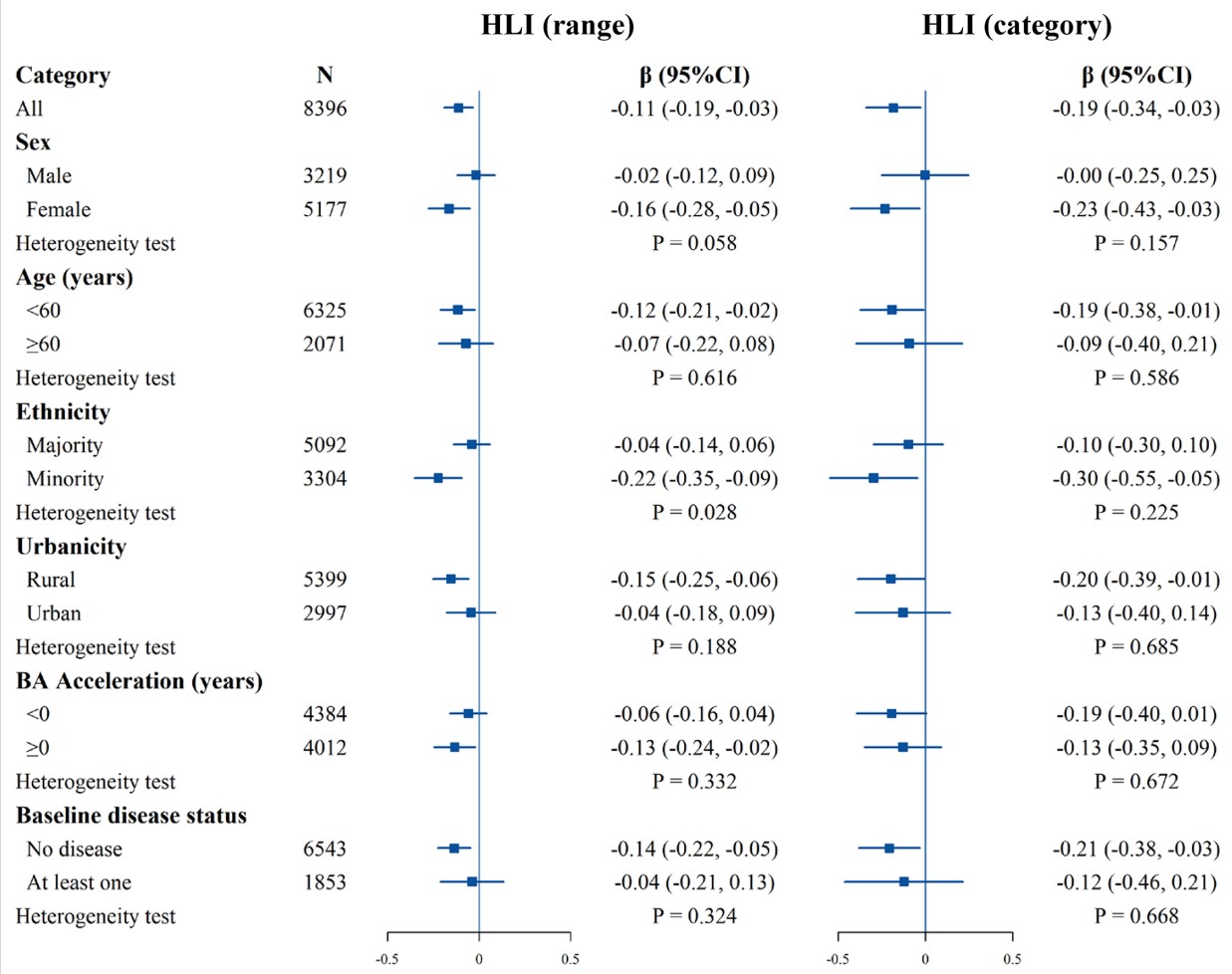

**Figure 4.** Stratified analysis of estimated associations between the healthy lifestyle index (HLI) and the comprehensive biological age (BA) acceleration. All models were adjusted for age, occupation, marital status, total energy intake, depression symptoms, anxiety symptoms, menopausal status in women, beverage intake, dietary supplement intake, diabetes, cardiovascular disease, cancer, sex, ethnicity, urbanicity, education, and the participants' age at baseline, with exclusion of the stratified variable as appropriate. The boxes represent point estimations. Horizontal lines represent 95% CI.

The online version of this article includes the following figure supplement(s) for figure 4:

**Figure supplement 1.** Stratified analysis of estimated associations between the healthy lifestyle index (HLI) and the cardiopulmonary biological age (BA) acceleration.

**Figure supplement 2.** Stratified analysis of estimated associations between the healthy lifestyle index (HLI) and the metabolic biological age (BA) acceleration.

**Figure supplement 3.** Stratified analysis of estimated associations between the healthy lifestyle index (HLI) and the liver biological age (BA) acceleration.

## Strengths and limitations

Unlike previous studies that used one measurement of lifestyle or BA, our study utilized longitudinal data from two waves of the CMEC study, which measured the longitudinal changes in lifestyle behaviors and BA of the body and multi-organ systems. We also assessed the impact of alterations in healthy lifestyle factors on validated BAs and further characterized the relative contributions. This comprehensive approach offers the latest evidence for the early prevention and targeted behavioral interventions of organ-specific aging and related diseases in southwestern China.

This study has certain limitations. Firstly, our construction of multi-organ BA was limited by the clinical lab data measures, preventing the development of BAs for some organ systems like the brain. Additionally, we could not fully capture specific aging processes of the immune system because of few immune-related indicators and the possible omission of critical markers. However, we used routinely

detected biomarkers to construct the BA, which are cost-effective and easily implemented. Therefore, this multi-organ BAs are suitable for broad adoption in primary prevention, especially in developing areas like Southwestern China. Besides, the CMEC had a limited range of diseases collected, precluding all BAs validation. Thus, we finally included only those BAs that could be validated for further analysis. Secondly, as an observational study, we established a prospective association between healthy lifestyles and accelerated aging, yet it does not confirm a causal relationship. Thirdly, the CMEC was established in 2018 with only data from two survey waves, which highlights the drawback of reduced statistical power of FEM. Future research could build on long-term, multi-wave follow-ups, employing FEM analysis based on repeatedly collected data to derive more reliable causal effect estimates. Fourthly, there is no gold standard for measuring biological aging. However, the KDM-BA method used in this study has the advantage of accurately predicting aging-related outcomes, and it has been validated in our study population (*Xiang et al., 2024*). Fifth, assessment of lifestyle factors was based on self-reported data collected through questionnaires, which may be subject to recall bias. Lastly, our study is based on a population from the Southwestern China, which could limit the generalizability of our findings. However, the Southwestern China is characterized by its diverse ethnicities and lower developmental level, with few studies addressing biological aging there. This research may provide crucial information for interventions in other less-developed regions.

## Conclusions

Within the Southwest China population, we observed that healthy lifestyle changes were inversely related to comprehensive and organ-specific accelerated biological aging, with diet and smoking making the most significant contributions to the comprehensive BA and metabolic BA separately. These findings underscore the potential for lifestyle interventions to slow the aging pace and identify priority intervention targets to various organ-specific aging in less-developed regions.

## Acknowledgements

We sincerely thank all the participants and staff of the CMEC study. We gratefully acknowledge Professor Xiaosong Li, the former principal investigator of CMEC research, for his leadership and tremendous contribution to the establishment of CMEC. Professor Li passed away in 2019.

## Additional information

### Competing interests

Xiong Xiao, Xing Zhao: A patent related to the CMEC study exists ('Risk Prediction Method for Metabolic Syndrome Based on SMOTE Technology and Random Forest Algorithm', Patent No. ZL 2021 1 0628911.9), on which two of our authors (Xiong Xiao and Xing Zhao) are co-inventors. The patent, held jointly by CMEC and Sichuan University, describes methods for predicting metabolic disease risk using CMEC data. While our current research draws from the same dataset, we investigated different research questions using distinct analytical approaches. We confirm that the intellectual property covered by this patent did not influence our study design, analysis methods, or the conclusions presented here. The other authors declare that no competing interests exist.

### Funding

| Funder | Grant reference number | Author |
|---|---|---|
| National Natural Science Foundation of China | 82273740 | Xiong Xiao |
| Natural Science Foundation of Sichuan Province | 2024NSFSC0552 | Xiong Xiao |

The funders had no role in study design, data collection and interpretation, or the decision to submit the work for publication.

## Author contributions

Yuan Zhang, Conceptualization, Formal analysis, Investigation, Writing - original draft; Dan Tang, Conceptualization, Formal analysis, Investigation, Methodology, Writing - original draft; Ning Zhang, Yi Xiang, Jianzhong Yin, Conceptualization, Supervision, Investigation, Writing – review and editing; Yifan Hu, Wen Qian, Yangji Baima, Xianbin Ding, Ziyun Wang, Supervision, Investigation, Writing – review and editing; Xiong Xiao, Conceptualization, Supervision, Funding acquisition, Investigation, Methodology, Writing – review and editing; Xing Zhao, Conceptualization, Supervision, Methodology, Writing – review and editing

## Author ORCIDs

Jianzhong Yin ◉ https://orcid.org/0000-0002-1876-387X
Xiong Xiao ◉ https://orcid.org/0000-0003-4471-7946
Xing Zhao ◉ https://orcid.org/0000-0001-5713-3603

## Ethics

This study was approved by the Sichuan University Medical Ethical Review Board [ID: K2016038, K2020022]. All procedures in the study were consistent with the 1964 Declaration of Helsinki and its subsequent amendments. All subjects agreeing to take part in the study signed informed written consent.

Reviewer #1 (Public review): https://doi.org/10.7554/eLife.99924.3.sa1
Author response https://doi.org/10.7554/eLife.99924.3.sa2

# Additional files

## Supplementary files

Supplementary file 1. Supplementary methods, descriptive analysis of biological age, and results of sensitivity analysis. (**a**) Detailed definitions of lifestyle factors in the main analysis and sensitivity analysis. (**b**) Candidate indicators used to construct the comprehensive and multi-organ systems of biological age. (**c**) Description of BA and BA acceleration. (**d**) Associations of the comprehensive and multi-organ systems BA acceleration with organ-specific diseases. (**e**) Associations of healthy lifestyle factors and HLI with the BA accelerations with adjustment for time-varying covariates. (**f**) Associations of healthy lifestyle factors and HLI with the BA accelerations altered healthy lifestyle criteria. (**g**) Associations of healthy lifestyle factors and HLI with the BA accelerations with additionally adjusted for BMI.

MDAR checklist

## Data availability

The research data is sourced from the China Multi-Ethnic Cohort (CMEC) Study, which is supported by national key R&D funding (Grant No. 2017YFC0907300). According to the funding agreement, all data from this cohort has been uploaded to the National Population Health Data Center (NPHDC; more information can be found at: https://www.ncmi.cn/phda/projectDataDetail.html?id=df7bdf3f-245d-3785-8552-47cb4c20e29c). For more detailed information, please contact CMEC Committee (sw_china_cohort@163.com). The de-identified data used in this study is publicly available via the Open Science Framework (https://osf.io/mqe6a/). The codes are provided at Lifestyles-and-biological-aging (https://github.com/Mitzi-zy/Lifestyles-and-biological-aging, copy archived at *Mitzi-zy, 2024*) and the processed data used to generate figures have been uploaded as the source data files.

The following dataset was generated:

| Author(s) | Year | Dataset title | Dataset URL | Database and Identifier |
| --- | --- | --- | --- | --- |
| Zhang Y | 2025 | Lifestyles-and-biological-aging | https://osf.io/mqe6a/ | Open Science Framework, mqe6a |

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
