## [Editor Report · eLife Assessment]

This **useful** study examined the associations of a healthy lifestyle with comprehensive and organ-specific biological ages defined using common blood biomarkers and body measures. Its large sample size, longitudinal design, and robust statistical analysis provide **solid** support for the findings, which will be of interest to epidemiologists and clinicians.

---

## [Referee Report · Reviewer #1 (Public review)]

Summary:

This study examined the associations of healthy lifestyles with comprehensive and organ-specific biological ages. It emphasized the importance of lifestyle factors in determining biological ages, which were using common blood biomarkers and body measures.

Strengths:

The data were from a large cohort study and defined comprehensive and six-specified BA.

Weaknesses highlighted previously:

(1) Since only 8.5% of participants from the CMEC were included in the study, has any section bias happened?

(2) The author should specify the efficiency of FFQ. How FFQ can genuinely reflect the actual intake? Moreover, how was the aMED calculated in your study?

(3) HLI (range) and HLI (category) should be clearly defined.

(4) The rationale of comprehensive and specific BA construction should be clearly defined and discussed. For example, can cardiopulmonary BA be reflected only by using cardiopulmonary status? I do not think so.

(5) The lifestyle index is defined based on an equal-weight approach, but this does not reflect reality and can not fully answer the research questions it raises.

Comments on the revised version:

The author answered most of the questions raised. However, since wine is the most important component of aMED, removing wine or alcohol may result in biased estimates. In addition, The authors acknowledge the limitations of this approach, namely that some biomarkers may not fully capture the complete aging process of the system; this weakness is particularly remarkable in organ-specific BA. The authors emphasize that it is cost-effective and easy to implement. However, the results associated with organ-specific BA may not be credible because they do not fully reflect the state of a particular organ. It is recommended that these shortcomings and the applicability of the results should be discussed in the text.

---

## [Author Response]

The following is the authors’ response to the original reviews.

**eLife Assessment**
This useful study examined the associations of a healthy lifestyle with comprehensive and organ-specific biological ages defined using common blood biomarkers and body measures. Its large sample size, longitudinal design, and robust statistical analysis provide solid support for the findings, which will be of interest to epidemiologists and clinicians.

Thank you very much for your thoughtful review of our manuscript. Your valuable comments have greatly helped us improve our manuscript. We have carefully considered all the comments and suggestions made by the reviewers and have revised them to address each point. Below, we provide detailed responses to each of the reviewers' comments. Please note that the line numbers mentioned in the following responses correspond to the line numbers in the clean version of the manuscript.

**Public Reviews:**

**Reviewer #1 (Public Review):**
Summary:This study was to examine the associations of a healthy lifestyle with comprehensive and organ-specific biological ages. It emphasized the importance of lifestyle factors in biological ages, which were defined using common blood biomarkers and body measures.Strengths:The data were from a large cohort study and defined comprehensive and six-specified biological ages.Weaknesses:(1) Since only 8.5% of participants from the CMEC (China Multi-Ethnic Cohort Study) were included in the study, has any section bias happened?

Thank you for your valuable question. We understand the concern regarding the potential selection bias due to only 8.5% of participants being included in the study. The baseline survey of China Multi-Ethnic Cohort Study (CMEC) employed a rigorous multi-stage stratified cluster sampling method and the repeat survey reevaluated approximately 10% of baseline participants through community-based cluster random sampling. Therefore, the sample of the repeat survey is representative. The second reason for the loss of sample size was the availability of biomarkers for BA calculation. We have compared characteristic of the overall population, the population included in and excluded from this study. Most characteristics were similar, but participants included in this study showed better in some health-related variables, one potential reason is healthier individuals were more likely to complete the follow-up survey. In conclusion, we believe that the impact of selection bias is limited.

**Author response table 1. sa2table1:** Baseline characteristics of participants included and not included in the study.

Characteristic	Overall N=99,556	Population included N=8396	Population not included N=91.160
Age (years)	50.48 [43.12, 60.27]	50.64 [43.91, 59.26]	50.46 [43.04, 60.38]
Female (%)	59762 (60.0)	5177 (61.7)	54585 (59.9)
Urbanicity (%)	33578 (34.1)	2997 (35.7)	30581 (33.9)
Majority (%)	55420 (56.2)	5092 (60.6)	50328 (55.8)
Education(%)			
No schooling	27028 (27.4)	2159 (25.7)	24869 (27.6)
Primary school	25136 (25.5)	2026 (24.1)	23110 (25.6)
Middle/high school	35850 (36.4)	3289 (39.2)	32561 (36.1)
College'university	10536 (10.7)	922 (11.0)	9614 (10.7)
Occupation(%)			
Primary industry practitioner	33842 (34.4)	2813 (33.5)	31029 (34.4)
Secondary industry practitioner	7002 (7.1)	532 (6.3)	6470 (7.2)
Tertiary industry practitioner	36792 (37.4)	3370 (40.1)	33422 (37.1)
Unemployed	20838 (21.2)	1681 (20.0)	19157 (21.3)
Married (%)	87420 (88.7)	7507 (89.4)	79913 (88.6)
Total energy intake (kcal/day)	1737.19[1354.75 2201.42]	1778.33 [1400.29, 2234.39]	1733.40[1350.15 2198.22]
BMI (kg/m^2^)	24.06 [21.78, 26.32]	24.22 [21.97, 26.67]	24.04 [21.76, 26.31]
Depression symptom (%)	4990 (5.1)	406 (4.8)	4584 (5.1)
Anxiety symptom (%)	5833 (5.9)	481 (5.7)	5352 (6.0)
Menopausal status in women (%)			
Premenopausal	26630 (45.0)	2378 (45.9)	24252 (44.9)
Perimenopausal	3994 (6.8)	359 (6.9)	3635 (6.7)
Postmenopausal	28544 (48.2)	2440 (47.1)	26104 (48.3)
Beverage (%)			
Never	90684 (92.1)	7856 (93.6)	82828 (91.9)
Former consumer	429 (0.4)	39 (0.5)	390 (0.4)
Currently consuming	7400 (7.5)	501 (6.0)	6899 (7.7)
Dietary supplement intake (%)	15679 (16.0)	1433 (17.1)	14246 (15.8)
HLI (healthy) (%)[Table-fn appR1table1fn3]	37104 (37.9)	3461 (41.2)	33643 (37.5)
**Major diseases**			
Diabetes (%)	4587 (4.7)	406 (4.8)	4181 (4.6)
CVD (%)	18226 (18.5)	1602 (19.1)	16624 (18.4)
Cancer (%)	782 (0.8)	67 (0.8)	715 (0.8)
**Biological ages** [Table-fn appR1table1fn4]			
Comprehensive BA	51.24 [42.20, 60.97]	51.08 [42.28, 60.03]	51.26 [42.19, 61.09]
Comprehensive BA acceleration	-0.15 [-3.15, 3.03]	-0.23[-3.28,2.88]	-0.14 [-3.14, 3.05]
Cardiopulmonary BA	50.76 [43.46, 60.34]	50.66 [43.87, 59.10]	50.77 [43.41, 60.45]
Cardiopulmonary BA acceleration	0.02 [-0.57, 0.57]	0.03 [-0.54, 0.58]	0.02 [-0.57, 0.57]
Metabolic BA	51.41 [41.91, 61.32]	51.57 [42.37, 60.89]	51.39 [41.86, 61.37]
Metabolic BA acceleration	-0.11 [-4.31, 4.14]	0.22 [-4.08, 4.44]	-0.14 [-4.33, 4.11]
Liver BA	51.88 [39.54, 63.59]	51.30 [39.49, 62.68]	51.94 [39.55, 63.69]
Liver BA acceleration	-0.12[-8.05, 7.94]	-0.37[-8.36, 7.73]	-0.10 [-8.03, 7.96]

BA, biological age; BMI, body mass index; CVD, cardiovascular disease; HLI, healthy lifestyle indicator.

Data are presented as median (25th, 75th percentile) for continuous variables and count (percentage) for categorical variables.

*For HLI, "healthy" corresponds to a score of 4-5.

†Information on each validated BA has been reported. BA acceleration is the difference between each BA and CA in the same survey.

BA, biological age; BMI, body mass index; CVD, cardiovascular disease; HLI, healthy lifestyle indicator.

1 Data are presented as median (25th, 75th percentile) for continuous variables and count (percentage) for categorical variables.

2 For HLI, "healthy" corresponds to a score of 4-5.

3 Information on each validated BA has been reported. BA acceleration is the difference between each BA and CA in the same survey.

(2) The authors should specify the efficiency of FFQ. How can FFQ genuinely reflect the actual intake? Moreover, how was the aMED calculated?

Thank you for the comments and questions. We appreciate the opportunity to clarify these aspects of our study. For the first question, we evaluated the FFQ's reproducibility and validity by conducting repeated FFQs and 24-hour dietary recalls at the baseline survey. Intraclass correlation coefficients (ICC) for reproducibility ranged from 0.15 for fresh vegetables to 0.67 for alcohol, while deattenuated Spearman rank correlations for validity ranged from 0.10 for soybean products to 0.66 for rice. More details are provided in our previous study (Lancet Reg Health West Pac, 2021). We have added the corresponding content in both the main text and the supplementary materials.

Methods, Page 8, lines 145-146: “The FFQ's reproducibility and validity were evaluated by conducting repeated FFQs and 24-hour dietary recalls.”

Supplementary methods, Dietary assessment: “We evaluated the FFQ's reproducibility and validity by conducting repeated FFQs and 24-hour dietary recalls. Intraclass correlation coefficients for reproducibility ranged from 0.15 for fresh vegetables to 0.67 for alcohol, while deattenuated Spearman rank correlations for validity ranged from 0.10 for soybean products to 0.66 for rice.”

For the second question, we apologize for any confusion. To avoid taking up too much space in the main text, we decided not to include the detailed aMED calculation (as described in Circulation, 2009) there and instead placed it in the supplementary materials:

“Our calculated aMED score incorporates eight components: vegetables, legumes, fruits, whole grains, fish, the ratio of monounsaturated fatty acids (MUFA) to saturated fatty acids (SFA), red and processed meats, and alcohol. Each component's consumption was divided into sex-specific quintiles. Scores ranging from 1 to 5 were assigned based on quintile rankings to each component, except for red and processed meats and alcohol, for which the scoring was inverted. The alcohol criteria for the aMED was defined as moderate consumption. Since the healthy lifestyle index (HLI) already contained a drinking component, we removed the drinking item in the aMED, which had a score range of 7-35 with a higher score reflecting better adherence to the overall Mediterranean dietary pattern. We defined individuals with aMED scores ≥ population median as healthy diets.”

Reference:

(1) Xiao X, Qin Z, Lv X, Dai Y, Ciren Z, Yangla Y, et al. Dietary patterns and cardiometabolic risks in diverse less-developed ethnic minority regions: results from the China Multi-Ethnic Cohort (CMEC) Study. Lancet Reg Health West Pac. 2021;15:100252. doi: 10.1016/j.lanwpc.2021.100252.

(2) Fung TT, Rexrode KM, Mantzoros CS, Manson JE, Willett WC, Hu FB. Mediterranean diet and incidence of and mortality from coronary heart disease and stroke in women. Circulation. 2009;119(8):1093-100. doi: 10.1161/circulationaha.108.816736.

(3) HLI (range) and HLI (category) should be clearly defined.

Thank you for the comment. We have added the definition of HLI (range) and HLI (category) in the methods section:

Methods P9 lines 165-170: “The HLI was calculated by directly adding up the five lifestyle scores, ranging from 0-5, with a higher score representing an overall healthier lifestyle, denoted as HLI (range) in the following text. We then transformed HLI into a dichotomous variable in this study, denoted as HLI (category), where a score of 4-5 for HLI was considered a healthy lifestyle, and a score of 0-3 was considered an unfavorable lifestyle that could be improved.”

(4) The comprehensive rationale and each specific BA construction should be clearly defined and discussed. For example, can cardiopulmonary BA be reflected only by using cardiopulmonary status? I do not think so.

Thank you for the opportunity to clarify. We constructed the comprehensive BA based on all the available biochemical data from the CMEC study, selecting aging-related markers (J Gerontol A Biol Sci Med Sci, 2021), and further construct organ-specific BAs based on these selected biomarkers. The KDM algorithm does not specify biomarker types but requires them to be correlated with chronological age (CA) (Ageing Dev, 2006). Existing studies typically construct BA based on available biomarker, we included 15 biomarkers in this study, which could be considered comprehensive and extensive compared to previous research (J Transl Med. 2023; J Am Heart Assoc. 2024; Nat Cardiovasc Res. 2024). For how the biomarkers for each organ-specific BAs were selected, we categorized biomarkers primarily based on their relevance to the structure and function of each organ system according to the classification in previous studies (Nat Med, 2023; Cell Rep, 2022). Since the biomarkers we used came from clinical-lab data sets, they were categorized based on the clinical interpretation of blood chemistry tests following the methods outlined in the two referenced papers (Nat Med, 2023; Cell Rep, 2022). We only used biomarkers directly related to each specific system to minimize overlap between the indicators used for different BAs, thereby preserving the distinctiveness of organ-specific BAs. We acknowledge the limitations of this approach that a few biomarkers may not fully capture the complete aging process of a system, and certain indicators may be missing due to data constraints. However, the multi-organ BAs we constructed are cost-effective, easy to implement, and have been validated, making them valuable despite the limitations.

Reference:

(1) Verschoor CP, Belsky DW, Ma J, Cohen AA, Griffith LE, Raina P. Comparing Biological Age Estimates Using Domain-Specific Measures From the Canadian Longitudinal Study on Aging. J Gerontol A Biol Sci Med Sci. 2021;76(2):187-94. doi: 10.1093/gerona/glaa151.

(2) Klemera P, Doubal S. A new approach to the concept and computation of biological age. Mech Ageing Dev. 2006;127(3):240-8. doi: 10.1016/j.mad.2005.10.004

(3) Zhang R, Wu M, Zhang W, Liu X, Pu J, Wei T, et al. Association between life's essential 8 and biological ageing among US adults. J Transl Med. 2023;21(1):622. doi: 10.1186/s12967-023-04495-8.

(4) Forrester SN, Baek J, Hou L, Roger V, Kiefe CI. A Comparison of 5 Measures of Accelerated Biological Aging and Their Association With Incident Cardiovascular Disease: The CARDIA Study. J Am Heart Assoc. 2024;13(8):e032847. doi: 10.1161/jaha.123.032847.

(5) Jiang M, Tian S, Liu S, Wang Y, Guo X, Huang T, Lin X, Belsky DW, Baccarelli AA, Gao X. Accelerated biological aging elevates the risk of cardiometabolic multimorbidity and mortality. Nat Cardiovasc Res. 2024;3(3):332-42. doi: 10.1038/s44161-024-00438-8.

(6) Tian YE, Cropley V, Maier AB, Lautenschlager NT, Breakspear M, Zalesky A. Heterogeneous aging across multiple organ systems and prediction of chronic disease and mortality. Nat Med. 2023;29(5):1221-31. doi: 10.1038/s41591-023-02296-6.

(7) Nie C, Li Y, Li R, Yan Y, Zhang D, Li T, et al. Distinct biological ages of organs and systems identified from a multi-omics study. Cell Rep. 2022;38(10):110459. doi: 10.1016/j.celrep.2022.110459.

(5) The lifestyle index is defined based on an equal-weight approach, but this does not reflect reality and cannot fully answer the research questions it raises.

Thank you very much for your valuable suggestion. We used equal weight healthy lifestyle index (HLI) partly to facilitate comparisons with other studies. The equal-weight approach to construct the HLI is commonly used in current research (Bmj, 2021; Diabetes Care. 2022; Arch Gerontol Geriatr. 2022). The equal-weight HLI can demonstrate the average benefit of adopting each additional healthy lifestyle and avoid assumptions about the relative importance of different behaviors, which may vary depending on the population. To further clarify the importance of each lifestyle factor, we conducted quantile G-computation analysis, which can reflect the weight differences between lifestyle factors (PLoS Med, 2020; Clin Epigenetics, 2022).

Reference:

(1) Zhang YB, Chen C, Pan XF, Guo J, Li Y, Franco OH, Liu G, Pan A. Associations of healthy lifestyle and socioeconomic status with mortality and incident cardiovascular disease: two prospective cohort studies. Bmj. 2021;373:n604. doi: 10.1136/bmj.n604.

(2) Han H, Cao Y, Feng C, Zheng Y, Dhana K, Zhu S, Shang C, Yuan C, Zong G. Association of a Healthy Lifestyle With All-Cause and Cause-Specific Mortality Among Individuals With Type 2 Diabetes: A Prospective Study in UK Biobank. Diabetes Care. 2022;45(2):319-29. doi: 10.2337/dc21-1512.

(3) Jin S, Li C, Cao X, Chen C, Ye Z, Liu Z. Association of lifestyle with mortality and the mediating role of aging among older adults in China. Arch Gerontol Geriatr. 2022;98:104559. doi: 10.1016/j.archger.2021.104559.

(4) Chudasama YV, Khunti K, Gillies CL, Dhalwani NN, Davies MJ, Yates T, Zaccardi F. Healthy lifestyle and life expectancy in people with multimorbidity in the UK Biobank: A longitudinal cohort study. PLoS Med. 2020;17(9):e1003332. doi: 10.1371/journal.pmed.1003332.

(5) Kim K, Zheng Y, Joyce BT, Jiang H, Greenland P, Jacobs DR, Jr., et al. Relative contributions of six lifestyle- and health-related exposures to epigenetic aging: the Coronary Artery Risk Development in Young Adults (CARDIA) Study. Clin Epigenetics. 2022;14(1):85. doi: 10.1186/s13148-022-01304-9.

**Reviewer #2 (Public Review):**
This interesting study focuses on the association between lifestyle factors and comprehensive and organ-specific biological aging in a multi-ethnic cohort from Southwest China. It stands out for its large sample size, longitudinal design, and robust statistical analysis.Some issues deserve clarification to enhance this paper:(1) How were the biochemical indicators for organ-specific biological ages chosen, and are these indicators appropriate? Additionally, a more detailed description of the multi-organ biological ages should be provided to help understand the distribution and characteristics of BAs.

We thank you for raising this point. As explained in our response to the fourth question from the first reviewer, we constructed the comprehensive BA b ased on all the available biochemical data from the CMEC study, selecting aging-related markers (J Gerontol A Biol Sci Med Sci, 2021), and further construct organ-specific BAs based on these selected biomarkers. The KDM algorithm does not specify biomarker types but requires them to be correlated with chronological age (CA) (Ageing Dev, 2006). Existing studies typically construct BA based on available biomarker, we included 15 biomarkers in this study, which could be considered comprehensive and extensive compared to previous research (J Transl Med. 2023; J Am Heart Assoc. 2024; Nat Cardiovasc Res. 2024). For how the biomarkers for each organ-specific BAs were selected, we categorized biomarkers primarily based on their relevance to the structure and function of each organ system according to the classification in previous studies (Nat Med, 2023; Cell Rep, 2022). Since the biomarkers we used came from clinical-lab data sets, they were categorized based on the clinical interpretation of blood chemistry tests (Nat Med, 2023). We only used biomarkers directly related to each specific system to minimize overlap between the indicators used for different BAs, thereby preserving the distinctiveness of organ-specific BAs.

We have added a descriptive table for the comprehensive and organ systems BAs in the supplementary materials to provide a more detailed understanding of the distribution and characteristics of BAs:

**Author response table 2. sa2table2:** Description of BA and BA acceleration.

BA	Comprehensive BA	Cardiopulmonary BA	Metabolic BA	Liver BA	Renal BA	Immune BA
**KDM-BA**						
Baseline	51.66 (12.45)	51.66 (11.50)	51.66 (13.31)	51.66 (17.35)	51.66 (15.24)	51.66 (14.00)
Follow-up	53.85 (12.04)	53.43 (10.84)	53.94 (13.01)	56.58 (17.59)	53.59 (14.67)	54.24 (13.51)
Change	2.49 (4.09)	2.00 (0.94)	2.32 (5.26)	5.09 (12.57)	2.31 (9.11)	2.68 (5.07)
**KDM-BA acceleration**						
Baseline	0.00 (4.83)	0.00 (0.91)	–0.00 (6.76)	–0.00 (13.02)	0.00 (10.03)	0.00 (8.04)
Follow-up	0.39 (4.89)	–0.04 (0.94)	0.48 (6.78)	3.11 (13.98)	0.13 (9.85)	0.77 (7.95)
Change	0.44 (4.09)	–0.05 (0.86)	0.27 (5.25)	3.05 (12.55)	0.26 (9.14)	0.63 (5.06)

BA, biological age.

Data are presented as mean (standard deviation).

(2) The authors categorized the HLI score into a dichotomous variable, which may cause a loss of information. How did the authors address this potential issue?

Thank you for raising this concern. We categorized each lifestyle factor into a binary variable based on relevant guidelines and studies, which recommend assigning a score of 1 if the guideline or study recommendations are met (Bmj, 2021; J Am Heart Assoc, 2023). While dichotomization may lead to some loss of information, it allows for a clearer interpretation and comparison of adherence to ideal healthy lifestyle behaviors. Another advantage of this treatment is that it allows for easy comparison with other studies. We categorized the HLI score into a dichotomous variable to enhance the practical relevance of the results (J Gerontol A Biol Sci Med Sci, 2021). Additionally, we conducted analyses using the continuous HLI score to ensure that our findings were robust, and the results were consistent with those obtained using the dichotomous HLI.

Reference:

(1) Verschoor CP, Belsky DW, Ma J, Cohen AA, Griffith LE, Raina P. Comparing Biological Age Estimates Using Domain-Specific Measures From the Canadian Longitudinal Study on Aging. J Gerontol A Biol Sci Med Sci. 2021;76(2):187-94. doi: 10.1093/gerona/glaa151.

(2) Klemera P, Doubal S. A new approach to the concept and computation of biological age. Mech Ageing Dev. 2006;127(3):240-8. doi: 10.1016/j.mad.2005.10.004

(3) Zhang R, Wu M, Zhang W, Liu X, Pu J, Wei T, et al. Association between life's essential 8 and biological ageing among US adults. J Transl Med. 2023;21(1):622. doi: 10.1186/s12967-023-04495-8.

(4) Forrester SN, Baek J, Hou L, Roger V, Kiefe CI. A Comparison of 5 Measures of Accelerated Biological Aging and Their Association With Incident Cardiovascular Disease: The CARDIA Study. J Am Heart Assoc. 2024;13(8):e032847. doi: 10.1161/jaha.123.032847.

(5) Jiang M, Tian S, Liu S, Wang Y, Guo X, Huang T, Lin X, Belsky DW, Baccarelli AA, Gao X. Accelerated biological aging elevates the risk of cardiometabolic multimorbidity and mortality. Nat Cardiovasc Res. 2024;3(3):332-42. doi: 10.1038/s44161-024-00438-8.

(6) Tian YE, Cropley V, Maier AB, Lautenschlager NT, Breakspear M, Zalesky A. Heterogeneous aging across multiple organ systems and prediction of chronic disease and mortality. Nat Med. 2023;29(5):1221-31. doi: 10.1038/s41591-023-02296-6.

(7) Nie C, Li Y, Li R, Yan Y, Zhang D, Li T, et al. Distinct biological ages of organs and systems identified from a multi-omics study. Cell Rep. 2022;38(10):110459. doi: 10.1016/j.celrep.2022.110459.

(3) Because lifestyle data are self-reported, they may suffer from recall bias. This issue needs to be addressed in the limitations section.

Thank you for your valuable suggestion. We acknowledge that the use of self-reported lifestyle data in our study may introduce recall bias, potentially affecting the accuracy of the information collected. We have added the following statement to the limitations section of our manuscript:

Discussion, Page 22, lines 463-464: “Fifth, assessment of lifestyle factors was based on self-reported data collected through questionnaires, which may be subject to recall bias.”

(4) It should be clarified whether the adjusted CA is the baseline value of CA. Additionally, why did the authors choose models with additional adjustments for time-invariant variables as their primary analysis? This approach does not align with standard FEM analysis (Lines 261-263).

Thank you for the opportunity to clarify. We have changed the sentence to “baseline CA”. For the second question, in a standard fixed effects model (FEM), only time-varying variables are typically included. However, to enhance the flexibility of our models and account for potential variations in the association of time-invariant variables with CA, as has been commonly done in previous studies, we additionally adjusted for time-invariant variables and the baseline value of CA (BMC Med Res Methodol, 2024; Am J Clin Nutr, 2020). Moreover, sensitivity analyses using the standard FEM were conducted in this study, and robust results were obtained.

Reference:

(1) Tang D, Hu Y, Zhang N, Xiao X, Zhao X. Change analysis for intermediate disease markers in nutritional epidemiology: a causal inference perspective. BMC Med Res Methodol. 2024;24(1):49. doi: 10.1186/s12874-024-02167-9.

(2) Trichia E, Luben R, Khaw KT, Wareham NJ, Imamura F, Forouhi NG. The associations of longitudinal changes in consumption of total and types of dairy products and markers of metabolic risk and adiposity: findings from the European Investigation into Cancer and Nutrition (EPIC)-Norfolk study, United Kingdom. Am J Clin Nutr. 2020;111(5):1018-26. doi: 10.1093/ajcn/nqz335.

(5) How is the relative contribution calculated in the QGC analysis? The relative contribution of some lifestyle factors is not shown in Figure 2 and the supplementary figures, such as Supplementary Figure 7. These omissions should be explained.

Thanks for the questions. The QGC obtains causal relationships and estimates weights for each component, which has been widely used in epidemiological research. More details about QGC can be found in the supplementary methods. The reason some results are not displayed is that we assumed all healthy lifestyle changes would have a protective effect on BA acceleration. However, the effect size of some lifestyle factors did not align with this assumption and lacked statistical significance. Because positive and negative weights were calculated separately in QGC, with all positive weights summing to 1 and all negative weights summing to 1, these factors would have had large positive weights. To avoid potential misunderstandings, we chose not to include these results in the figures. We have added explanations to the figure legends where applicable:

“The blue bars represent results that are statistically significant in the FEM analysis, while the gray bars represent results in the FEM analysis that were not found to be statistically significant and positive weights were not shown.”

**Recommendations for the authors:**

**Reviewer #2 (Recommendations For The Authors):**
To enhance this paper, some issues deserve clarification:(1) How were the biochemical indicators for organ-specific biological ages chosen, and are these indicators appropriate? Additionally, please provide a more detailed description of the multi-organ biological ages to help understand BAs' the distribution and characteristics.(2) The authors categorized the HLI score into a dichotomous variable, which may cause a loss of information. How did the authors address this potential issue?(3) Because lifestyle data are self-reported, they may suffer from recall bias. This issue needs to be addressed in the limitations section.(4) Lines 261-263: Please clarify if the adjusted CA is the baseline value of CA. Additionally, why did you choose models with additional adjustments for time-invariant variables as your primary analysis? This approach does not align with standard FEM analysis.(5) How is the relative contribution calculated in the QGC analysis? The relative contribution of some lifestyle factors is not shown in Figure 2 and the supplementary figures, such as Supplementary Figure 7. Please explain these omissions.

The above five issues overlap with those raised by Reviewer #2 (Public Review). Please refer to the responses provided earlier.

Minor revision:Line 50: The expression "which factors" should be changed to "which lifestyle factor."

Thank you for the suggestion. As suggested, we have used “which lifestyle factor” instead.

Lines 91-92: "Aging exhibits variations across and with individuals" appears to be a clerical error. According to the context, it should be "Aging exhibits variations across and within individuals."

We thank the reviewer for the correction. We have updated the text to read:

“Aging exhibits variations across and within individuals.”

Line 154: The authors mentioned "Considering previous studies" but lacked references. Please add the appropriate citations.

Thank you for pointing this out. We apologize for the oversight. We have now added the appropriate citations to support the statement "Considering previous studies" in the revised manuscript.

Lines 170-171: "regular exercise ("12 times/week", "3-5 times/week," or "daily or almost every day")"; the first item in parentheses should be "1-2 times/week"? Please verify and correct if necessary. Additionally, check the entire text carefully to avoid confusion caused by clerical errors.

Thank you for your careful review. We have changed the sentence to "1-2 times/week." We have thoroughly checked the entire manuscript to ensure that no other clerical errors remain.

Clarifications for Table 1:i. The expression "HLI=0" is difficult to understand. Please provide a more straightforward explanation or rephrase it.

Thank you for your feedback. We have removed the confusing expression and provided a clearer explanation in the table legend for better understanding:

“For HLI (category), "healthy" corresponds to a score of 4-5, while "unfavorable" corresponds to a score of 0-3.”

ii. The baseline age is presented as an integer, but the follow-up age is not. Please clarify this discrepancy.

Thank you for pointing out this discrepancy. We calculated the precise chronological age based on based on participants' survey dates and birth dates for the biological age calculations. Initially, the table presented age as integers, but we have now updated it to show the precise ages.